# Shaping in the Third Direction; Fabrication of Hemispherical Micro-Concavity Array by Using Large Size Polystyrene Spheres as Template for Direct Self-Assembly of Small Size Silica Spheres

**DOI:** 10.3390/polym14112158

**Published:** 2022-05-26

**Authors:** Ion Sandu, Claudiu Teodor Fleaca, Florian Dumitrache, Bogdan Alexandru Sava, Iuliana Urzica, Iulia Antohe, Simona Brajnicov, Marius Dumitru

**Affiliations:** 1National Institute for Lasers, Plasma and Radiation Physics, Lasers Department, 409 Atomistilor Street, 077125 Bucharest, Romania; ion.sandu@inflpr.ro (I.S.); claudiu.fleaca@inflpr.ro (C.T.F.); florian.dumitrache@inflpr.ro (F.D.); iuliana.iordache@inflpr.ro (I.U.); iulia.antohe@inflpr.ro (I.A.); brajnicov.simona@inflpr.ro (S.B.); 2Faculty of Applied Chemistry and Materials Science, University Politehnica of Bucharest, 313 Splaiul Independenţei Street, Sector 6, 060042 Bucharest, Romania

**Keywords:** colloidal crystal, self-assembly, concavity, shape, super-structure

## Abstract

Silica and polystyrene spheres with a small size ratio (r = 0.005) form by sequential hanging drop self-assembly, a binary colloidal crystal through which calcination transforms in a silica-ordered concavity array. These arrays are capable of light Bragg diffraction and shape dependent optical phenomena, and they can be transformed into inverse-opal structures. Hierarchical 2D and 3D super-structures with ordered concavities as structural units were fabricated in this study.

## 1. Introduction

Colloidal solutions containing spheres of sub-micron dimensions can form through the phenomenon of self-assembly, following the loss of their liquid by evaporation, an ordered porous solid, known as colloidal crystal (CC) or artificial opal (AO) [1,2,3,4].

Colloidal crystals are usually self-assembled under gravity, electrostatic force, and capillary force [5]. The self-assembled spheres are weakly bonded [6], and this weak mechanical strength restricts their applications mainly to the theoretical study of the self-assembly phenomenon [7,8,9] and colloidal lithography [10,11] where, through a relatively simple approach, structured material arrays on nanometer-scale can be fabricated.

However, by filling the voids between spheres with a fluid precursor that is capable of solidification followed by sphere-removal, a new class of materials is obtained, namely inverted opals (IOs) [12,13,14]. With an increased mechanical strength, IOs preserve the shape and packing lattice of CCs, but instead of spheres, spherical holes that are connected through continuous channels fill a desirable material. By combining their ability to produce light diffraction and/or to allow the movement of fluids through their pores with some specific physical, chemical, or biological properties of functional materials, a large number of possible applications result: opto-chemical and opto-biological sensors [15,16] to detect chemical and living events, band-stop and bandpass filters [17,18] for integrated optics, micro-antennas [19] for satellite and mobile communications, laser components [20], catalysts [21,22], porous electrodes [23], membranes for smart filtration [24], scaffolds for cells and tissue growing [25], etc.; however, these are only a small part of them.

Many potential applications of IOs regard on-demand tuning of the Bragg reflection or transmission of light (Equation (1)) to visibly change the colors by external stimuli [3,26].
(1)$$m\lambda =2d_{hkl}\sqrt{{\left[n_{1}f+n_{2}\left(1-f\right)\right]}^{2}-sin^{2}}\theta$$
where *m* is the diffraction order, *λ* is the wavelength of the reflected (or transmitted light); *n*_1_, *n*_2_ are the refractive indices of IO solid material and of that which fills the voids respectively; *d*_*hkl*_ is the lattice period of the crystalline direction of the propagation of light; *f* is the filling factor, a parameter which strongly depends on the specific synthesis parameters and can take 0.74 as its maximum value; and *θ* is the angle between the incident beam and the normal-to-diffraction crystal planes.

At first glance, the wavelength *λ* of the reflected or transmitted light from Equation (1) is fixed by physical parameters such as the refractive indices of dielectric materials and the lattice constants of IO structures. The lattice itself can take several configurations depending on the CCs synthesis technique. A single layer of closely packed spheres presents a hexagonal or a square lattice and a 3D film can be seen as stacks of these single layers. There are three ways of stacking the layers: hexagonal, face-centered cubic (fcc), and double-hexagonal [27]. Synthesized inverted opals mainly self-assemble in a fcc lattice, thus the lattice period *d*, can be written as:(2)$$d_{hkl}=D\sqrt{\frac{2}{h^{2}+k^{2}+l^{2}}}$$
where *D* is the diameter of the spheres (or holes), and *h*, *k*, *l* are the Miller indices [28], thus for a light reflection peak position we can vary either the sphere’s size *d*, or the packing lattice (*h*, *k*, *l*).

However, the definition of *θ* in Equation (1) hides a very important parameter which could strongly modify the light diffraction in colloidal crystals: the crystal’s shape. In the case of simple three-dimensional figures, there are two classes of shapes: straight shapes, such as, rectangular, cubic, pyramidal, truncated pyramid, and prisms of all kinds which have faces and edges as attributes; and curved shapes, such as, spherical, hemispherical, cylindrical, conical, or toroidal which possess some specific attributes such as curvature and focus. One of the most important geometrical features which differentiates these shapes is the normal direction N, in each point of the surface (Figure 1a). For a straight shape, the normal directions on a facet have the same orientation which suddenly changes at the edge of a neighbor facet, while a round shape continuously changes the orientation of its normal direction; thus, they intersect in the center of the curvature. At a half of the distance between this point and the pole of the curved surface we find the focal point; a high level of optical phenomena (physical optics) are described through the focal point. Straight shaped crystals which present facets that are non-parallel with the substrate have their one normal direction different to that of the substrate and can introduce specific optical features. This geometry is known as asymmetrical Bragg diffraction [29]. Further, the curved shaped surfaces can induce beside wave (physical) optics and geometrical (ray) optics phenomena. For an easier management of surface phenomena, the substrate normal is usually taken as a reference and the measurements are reported to the substrate normal; thus, many times, the normal to the colloidal crystal is considered as the normal to the substrate. However, this is correct only for one kind of shaped colloidal crystal that is flat and parallel to the substrate surfaces.

Most parts of supported artificial opals (Figure 1b) and subsequently of inverted opals are flat and have a rectangular shape. They have extended sizes in the (x, y) plane, from square millimeters to square centimeters, and are only micrometers or a few tens of micrometers in thickness. Their upper surface is smooth, straight, and parallel with a flat substrate. Its normal to the upper surface is the same with the substrate normal direction (Figure 1b). In the great majority of cases, self-assembled colloidal crystals exhibit only (111) dense-packed planes at the surface [30] or in some cases, on the fcc (100) or bcc (100) [31], (001), or (110) planes [32]. The simultaneous controlled presence of multiple types of surface planes in these photonic crystals would greatly enrich their photonic bands and, consequently, their potential applications. The tuning and control of an increased complexity also depends on the CCs’ response intensity. The intensity of generated effects is proportional with stimuli intensity, the density to the surface of the active structures, and the area of the interaction zone (the spot area of the light beam, S). A plane that is flat and parallel with the substrate surface can support an extended interacting area, S, and is independent of the S localization; However, these considerations are not valid for other shapes.

According to Equation (1), straight shaped CCs show different structural colors when observed from different angles, resulting in numerous applications.

However, other applications require a wider viewing angle or, in contrast, an even greater dependence between *λ* and *θ*. They may even require that the CCs exhibit different optical features for different incident angles.

A sharper variation of the wavelength with the incident angle can be achieved on very thin colloidal-crystal films (single or bilayer) in transmitted light, for a transmitted angle that is different from the incident angle (non-Bragg diffraction) even when the normal to the substrate is the same with the normal to the film. This phenomenon is known as the super-prism effect [33,34] and it is not caused by the CCs’ shape. They are too thin to have a 3D shape.

Different optical features for different incident angles could be obtained by shaping CCs with flat surfaces but non-parallel with the substrate: a pyramid [35], a triangular prism [34], or a truncated pyramid [36,37] (Figure 1c), for example. In these cases, each CCs’ face shows a different diffraction plane family normal to the surface. Their spectra exhibit different characteristics depending on the orientation of the facet. However, this effect requires a minimum shaped surface and a light spot diameter that is at most equal with the shaped surface [36,37]. These conditions strongly minimize the phenomenon.

A weak dependence between the wavelength of reflected light *λ* and the incident angle *θ* is obtained in the case of curved colloidal crystals that are shaped as spheres [38,39], hemispheres (Figure 1d) [40,41], hemispherical cavities (Figure 1e), cylinders (Figure 1f) [42], or toroidal-like structures [43].

First, extended crystals with a straight lattice do not occur in a curved geometry. For this reason, ordered polygonal areas that are separated by amorphous boundaries will be formed. The smaller the radius of curvature, the higher the number of these crystalline islands and implicitly, the number of defects. Second, the weakly dependent effect between the wavelength of reflected light *λ* and the incident angle *θ* is a complex phenomenon given by the Bragg diffraction of light on the planes (111) and (220) of a fcc network that has oriented planes (111) parallel to the surface at each point of the sphere (Figure 1d,e) [8,44,45,46,47,48]. The phenomenon works for curved colloidal crystals with sizes between a few tens of micrometers and a few hundred micrometers. If the crystals are too small, they are likely to be disordered or to have too few (111) planes, but if they are too large, they tend to present an incompletely colored surface.

As we can see, both shape-dependent phenomena have severe size-dependent restrictions. These restrictions could be removed by the approach of patterning: instead of a small shaped colloidal crystal, an array of a huge number of shaped CCs. A patterned colloidal crystal is a discrete, hierarchically ordered structure of small and identical colloidal crystal units which preserve all the properties of a larger crystal. Furthermore, it has some collective properties (Figure 1g) [49,50,51,52,53]. Bragg diffraction works on a patterned crystal just like on a bulk one. However, by such an approach the size and the shape of the individual colloidal crystal monolith must be controlled, and this does not appear to be simple. Most of the patterned colloidal crystals reported so far present identical pattern units or a different structuration only in the (x, y) plane (Figure 1g), and only a few present different structuration along all of three space dimensions [50]. The use of more complex CC units with individual shapes enables even more sophisticated symmetries and structural features over multiple length scales. This is the case with so-called “superstructures” which are arrays or patterns of 3D shaped colloidal crystals units (Figure 1h). These superstructures can look like convex hemispherical colloidal-crystal arrays as a result of colloidal droplets printed onto a substrate [40,54]; spherical or toroidal-colloidal crystals produced by emulsion or microfluidic techniques arranged as monolayers [38,55]; triangular prisms formed as grids by using a polyester fabric as scaffold [34]; or aligned and packed cylinders in whose insides sub-micron spheres have crystallized [56].

In the present article we show how such a superstructure, namely an ordered hemispherical concavity array, can be obtained by using large-size polystyrene micrometric spheres as scaffolds for the self-assembly of small-size silica sub-micron spheres (Figure 1i). To use spheres of two different sizes for their self-assembly is a normality for an already mature scientific domain, namely Binary Colloidal Crystals [57,58,59]. However, in this domain, the limits for the spheres’ size ratio r that is defined as the ratio between small size spheres and large size spheres varies between 0.4 and 0.1. These limits allow a rich structuration of the colloidal crystal films in the x-y plane parallel to the substrate, but even when the smallest is used, namely 0.1, the spheres are not able to form a structured film in the z directions. This can be realized for much smaller ratios. In our case the ratio is 0.005, a few orders of magnitude lower, which allowed the obtaining of a quality colloidal crystal array structured in all three directions. Moreover, three-dimensional patterned colloidal crystals having as structural units ordered concavities (rather than simple colloidal crystals) as units can be constructed. This could be a new level of patterning.

## 2. Materials and Methods

### 2.1. Materials

Polystyrene (PS) sphere aqueous-colloidal solutions with a 20.04 and 50.7 µm mean diameter, 10% *w/v*; as well as sub-micron sphere aqueous-colloidal solutions of PS with a 0.488 µm mean diameter, 5% *w/v*; and of SiO_2_ with a 0.264 µm mean diameter, 5% *w/v*, were purchased from microParticles GmbH, Berlin, Germany and either used as they were or diluted with deionized water when needed. Commercially available microscope glass slides were used as substrates after a few minutes of cleaning by ultrasonication in acetone (min.99.6%, Silal Trading, Bucharest, Romania), distilled water (5μS/cm, prepared with a Dest-4 water distiller, from JP Selecta, Barcelona, Spain) and ethanol (min. 99.5%, Chimreactiv S.R.L., Bucharest, Romania), followed by their natural drying. Also aqueous hydrofluoric acid (48% Merk, Darmstadt, Germany), aqueous sodium silicate (50%, Silal Silal Trading, Bucharest, Romania) and tetraethyl orthotitanate (97%, Fluka, Buchs, Switzerland) were employed for some experiments.

### 2.2. The Synthesis of Large Size Sphere Single Layer

By using a microsyringe, a droplet of colloidal polystyrene solution (20.04 µm or 50.7 µm, 0.5–1.0% *w/v*) was deposited on the surface of a glass slide. The lamella was turned upside down and with the tip of the syringe in contact with the deposited hanging drop the volume of solution was supplemented up to 50 μL. The drop was allowed to evaporate under laboratory conditions for 1–2 h.

### 2.3. The Synthesis of SiO_2_ Hemispherical Micro-Concavity Array

By using a microsyringe, a droplet of colloidal SiO_2_ solution (0.264 µm, 5% *w/v* mixed with ethanol, 4/1 volume ratio) was gently deposited onto the PS large sphere single layer prepared, as above mentioned. After one minute, the lamella was turned upside down and the solution volume was supplemented up to 30 μL. The drop was allowed to evaporate under laboratory conditions for 1–2 h. The last step was to heat the sample at 500 °C for 30 min.

### 2.4. The Synthesis of Inverse Opals Hemispherical Micro-Concavity Array

#### 2.4.1. Sodium Silicate (Na_2_SiO_3_) Non-Structured Inverse Opals

A single layer that resulted from Section 2.2 was dipped in sodium silicate solution, 20% wt., for 30 min. It was taken out and left to dry naturally overnight. The last step was to heat the sample at 500 °C for 30 min.

#### 2.4.2. Sodium Silicate (Na_2_SiO_3_) Structured Inverse Opals

By using a microsyringe, a droplet of colloidal PS solution (0.488 µm, 5% *w/v* mixed with ethanol, 4/1 volume ratio) was gently deposited onto the single layer that resulted from Section 2.2. After one minute, the lamella was turned upside down and the solution volume was supplemented up to 30 μL. The drop was allowed to evaporate under laboratory conditions for 1–2 h. Then, it was dipped in sodium silicate solution, 20% wt., for 30 min. It was taken out and left to dry naturally overnight. The last step was to heat the sample at 500 °C for 30 min.

#### 2.4.3. Titanium Oxide (TiO_2_) Structured Inverse Opals

As in the case of sodium silicate (Na_2_SiO_3_) structured inverse opals but instead Na_2_SiO_3_, a solution of titanium dioxide precursor that was prepared from tetraethyl orthotitanate TiC_8_H_20_O_4_ (TEOT) 98% was solubilized in 12 mL ethanol (C_2_H_5_OH: TiC_8_H_20_O_4_ molar ratio 20:1) with the aid of a magnetic stirrer (400 rpm) for 30 min, followed by adding 0.25 mL HCl 0.1 N, after which it was stirred for another 30 min at 500 rpm.

#### 2.4.4. Polystyrene Structured Inverse Opals

Up to the thermal treatment the procedure was the same as in Section 2.3. However, instead of heating at 500 °C, the sample was kept in a furnace at 250 °C, for one hour. Sheets of scotch tape were stacked on the glass lamella, close to the surrounding interest zone. A few drops of aqueous hydrofluoric acid (25% wt.) were dropped onto the sample between the scotch tapes (their role is to not allow the HF to spread all over the lamella), allowing the acid to dissolve the SiO_2_ spheres (five minutes). The acid was periodically removed and refreshed three times. Then it was washed with distilled water and left to dry.

### 2.5. The Synthesis of Superstructures Based on Hemispherical Micro-Concavity

#### 2.5.1. Synthesis of SiO_2_/PS Hemispherical Micro-Concavity Array

One drop of PS colloidal solution (0.488 µm, 5% *w/v*, mixed with ethanol, 4/1 volume ratio) was deposited onto the SiO_2_ hemispherical micro-concavity array that was prepared in Section 2.3. The drop was allowed to evaporate under laboratory conditions for 1–2 h.

#### 2.5.2. Binary Hemispherical Micro-Concavity Array

PS colloidal solutions (50.7 and 20.04 µm) were mixed in a ratio of 1/3 and a large drop was deposited (sessile manner) onto the glass substrate. After drying, we proceeded as in Section 2.3.

#### 2.5.3. SiO_2_ Hemispherical Micro-Concavity Grid

A droplet of PS colloidal solutions (50.7 or 20.04 µm, 5% *w/v*) was deposited onto the glass substrate. A polyester fabric sheet (5 mm × 5 mm) was deposed onto the colloidal droplet and slightly tapped till the fabric sheet became wet and stuck to the glass substrate. Further (2–3) PS colloidal droplets were deposed onto the fabric surface. After liquid evaporation the fabric was removed and we proceeded to the end as in the Section 2.3.

### 2.6. Investigations

Macro-scale observations of the as-synthesized patterned colloidal crystals were performed using optical microscopy. A scanning electron microscope (SEM) (Apreo S Thermo Fisher Scientific, Auburn, AL, USA) was used to observe the structures and morphologies of the self-assembled patterned colloidal crystals. A thin layer of gold was sputtered onto the samples prior to imaging. UV–Vis transmittance and reflectance spectra were acquired by using optical fibers that were connected to an AvaLight-DHc light source (spot size~200 μm) and an AvaSpec-ULS2048CL-EVO, both high-resolution spectrometers from Avantes, Apeldoorn, The Netherlands.

## 3. Results and Discussion

The fundamental idea of this paper is to use a monolayer of polystyrene micrometric sphere array as a template for the direct self-assembly of sub-micron spheres. It is a perfect analogy for the classical synthesis of the inverse-opal in which a molecular solution is infiltrated in an artificial opal. If the smooth concavities are usually obtained at the submicron level, this time we expect to form concavities with the size of micrometric spheres whose inner walls are formed by arranged submicron spheres.

The main quality of the “Hanging drop” method of colloidal crystals synthesis [60] (Figure 2a) is the formation of high thickness CCs. However, we noticed that it can also form a monolayer of micrometric-sized PS spheres if a dilute colloidal solution (0.5–1.0% *w*/*v*) is used. The lack of frictional forces between the spheres and the substrate (the colloidal crystal film is formed at the liquid/air interface) [60] and the favorable orientation of gravity [60] result in a monolayer with large-ordered domains (tens of spheres) (Figure 2b). Due to the relatively large voids between the micrometric spheres, a molecular solution (Na_2_SiO_3_) infiltrates easily and after calcination a quality inverse opal results (Figure 2c). However, a colloidal solution of sub-micron spheres must not only homogeneously and completely infiltrate the opal, but also form a CC that is crystalized and structurally resistant to heat treatment.

A simple estimation of the silica spheres colloidal solution volume required for infiltration performed by using Equation (3):(3)$$V_{SiO2\;coll.\;}=\frac{1}{2}\frac{f\left(1-f\right)\pi D^{2}d_{PS}\rho_{SiO2}}{4c_{SiO2}}$$
where we consider a circular single-layer of close packed polystyrene spheres forming a flat cylinder whose voids are fully occupied by silica spheres (*f* being the same filling factor from Equation (1); *D* is the polystyrene sphere spot diameter; *d_PS_* is its height (equal with the diameter of polystyrene sphere); *ρ_SiO_*_2_ is the silica sub-micron particle density; and *c_SiO_*_2_ is the silica colloid concentration) it shows that a volume of 3 μL of silica colloidal solution is sufficient to fill the voids of a 7 mm diameter spot containing 20 μm polystyrene spheres. However, we encountered some experimental problems. Both the 20 μm and 50 μm sphere monolayers proved to be hydrophobic, the aqueous colloidal solution does not infiltrate between the large size PS spheres but remains above them. By adding an amount of up to 25% vol. of ethanol in the colloidal solution of SiO_2_ we managed to infiltrate these small spheres, but the colloidal solution spreads over a much larger area on contact with the substrate. Infiltration by the hanging drop method (Figure 2d) was imposed because in this case the ratio between drop volume and its contact surface is much higher than in the case of a sessile drop. Regardless, a part of the silica spheres self-assembles at the edge and outside the monolayer. By using a theoretical estimated volume of the silica colloidal solution, extremely defective structures were formed. By the gradual increase in the volume of infiltrated colloidal solution we obtained structures of the same height whose structural integrity increased proportionally with the increase in the colloid silica volume. This could happen if the crystallization took place from the surface of the polystyrene spheres to the interior of the volume between the spheres (low silica colloid volumes means thin silica walls) and any excess silica spheres were removed after heat treatment due to the melting of the polystyrene spheres (which, by increasing its volume removes from the system all excess silica spheres deposited above half of each polystyrene sphere). The sessile and hanging co-assembly of large PS and small silica spheres leads to the formation of a thick layer of silica spheres at the interface with the substrate and a non-close-packed array.

After the complete evaporation of the colloid’s solvent, we can observe (Figure 2e) that the array of PS spheres keeps its initial order, self-assembling in a polycrystalline structure with organized mono-domains (Figure 2f) that are large enough to induce the Bragg diffraction of light. Indeed, recordings of a UV–Vis transmission spectrum (inset Figure 2f white curve) show the existence of two transmission minima placed at *λ*_1_ = 587 nm and *λ*_2_ = 486 nm. The fact that we can observe both diffraction features indicates that the silica spheres were self-organized as a twin crystal [39,61]. By using Equations (1) and (2) we can calculate the theoretical band-gap positions. Silica spheres of *d* = 264 nm, *n* =1.45, *f* = 0.74, and diffraction planes (111) and (200) of an fcc lattice give λ_111_ = 573 nm and *λ*_200_ = 496 nm. These values were compared with those extracted from the UV-Vis transmission spectra of a 264 nm silica opal that was synthesized by the hanging drop method (a compact block opal) [60], that experimentally shows *λ*_111_ = 580 nm and *λ*_200_ = 486 nm. We see no remarkable difference between the transmission spectra of these silica spheres that are self-organized between PS spheres, and a classical close-packed silica spheres opal that was synthesized by us by the hanging drop method (inset Figure 2f). We could not identify a shape (concavity) effect that was induced through classical optical measurements.

After the removal of the polystyrene spheres by calcination, regular points that maintain the order of the polystyrene spheres in a monolayer could be seen at SEM (Figure 3a). A closer look shows a honeycomb network, a common image of inverse-opal structure (Figure 3b Each white dot in Figure 3a is a hemispherical micro-concavity (Figure 3b) with a hole of a few micrometers in diameter in the concavity center (the place where the polystyrene spheres were in contact with the substrate)(Figure 3c). If the concavities are obtained by co-assembly then this empty place is missing (Figure 3d). The diameter of the concavities that was obtained by sequential infiltration is very close to that of the polystyrene spheres that were used, and slightly smaller in the case of those that were obtained by co-assembly.

Because both atomic force microscopy (AFM) and the profilometry investigations on these concavities fails (due to AFM tip mismatching with the cavity sizes, and the low mechanical resistance of the structures, respectively), tilted (45°) SEM images (Figure 3e,f) show us the real hemispherical shape of ordered cavities formed by close-packed silica 264 nm spheres. The most common view on SEM was a hexagonal lattice with a length order which increased from the substrate (concavity bottom) to the top. The most extended order was found to the flat surface between the concavities, thus the “single crystal” size varied between 1–10 μm.

At the optical microscope (transmission), the film has a red color (Figure 3g), the light can pass through both the holes in the center of each concavity and the compact silica sub-micron spheres, and the UV-Vis spectrometry (Figure 3h) shows two transmission dips placed close to the same wavelengths, the same as was found before calcination. The optical microscopy reflectance images show an intense homogeneous green-yellow color (Figure 3i) and the concavities are perfectly dark. Reflectance UV-Vis spectrometry (Figure 3j) records two reflection maxima in the same places as the dips from the transmitted light. It seems that the patterned silica concavity array optically behaves the same as a classical close-packed, rectangular, silica opal. No shape induced optical phenomenon could be noticed.

However, taking optical microscopy images at various successive magnifications (Figure 4a–c, e–g), we noticed the existence of some light diffraction rings in the peripheral part of each concavity when an objective with a certain numerical aperture (NA = 0.50) was used. These diffraction rings are more intense in transmitted light (Figure 4b) and less intense in reflected light (Figure 4f). Angle-resolved UV-Vis spectrometry shows no dip displacement in transmitted light (Figure 4d) and a possible displacement of the band generated by the family of planes (111), as well as a stagnation of the band generated by the planes (200) in reflected light (Figure 4h) when the angle of incidence is varied from 0 degrees to 20 degrees (a characteristic of curved colloidal crystal [38,39,40,41,42,43]). However, this displacement is uncertain in the absence of the deconvolution of the reflectance spectra and of other complementary investigations, and it could explain the difference of the rings’ intensity (transmission and reflection), rather than their existence.

These diffraction rings in the peripheral part of the concavities seem to be a shape-dependent phenomenon which, although weak, could be constantly observed. Similar diffraction rings were reported in the case of photonic balls-colloidal crystals by R. Ohnuki et al., [62] when they were observed using objectives with different numerical apertures. They considered that the rings are caused by the tilted planes and refraction at the spherical surface which causes reflection in a highly oblique direction. Even if there are some significant differences between their photonic balls and ours superstructures such as: convex versus concave shaped crystals; the choice of silica sphere size; and the use of the (220) and (111) planes family instead of the (200) and (111) family, in our case, too many similarities remain to be studied, considering that we observed the same phenomenon.

A natural continuation of shaping colloidal crystals was to achieve an ordered micro-concavity inverse opal array. Proceeding in a classical manner (Figure 5a), an array composed of 20 μm or 50 μm PS spheres (among which 0.46 μm PS spheres were infiltrated and self-assembled) was immersed in liquid solutions of sodium silicate or titania precursor (see Section 2.4.2 and Section 2.4.3). After removal of the polystyrene spheres by calcination the infiltrated binary colloidal crystals were transformed into inverse opal structures (Figure 5b,c).

However, though their quality is low and both the Na_2_SiO_3_ and the TiO_2_ precursor produce structures with many defects, they still generate the Bragg diffraction of light (Figure 5d). These extensive defects are due to the incomplete infiltration of precursor solutions, heat treatment inducing the shrinking phenomenon associated with multiple cracking generation, as well as the inherent appearance of an overlayer, a frequent phenomenon that also occurs in the case of classical rectangular block opals and is difficult to cancel. However, during the experiments we noticed the possibility of obtaining polystyrene inverse-opals where the infiltrated material between the silica spheres derives from the large polystyrene spheres via melt infiltration from inside the film, as suggested in Figure 5e. We found that an array of PS and silica spheres (such as the one in Figure 2) that are kept in the oven at a temperature at which the polystyrene becomes liquid turn into an array of cavities with polystyrene that is completely infiltrated among the silica spheres (Figure 5f–h). High quality inverse opals resulted from the dissolution of the silica spheres with aqueous HF (Figure 5i). Through these structures the light can be transmitted (Figure 5j) or reflected (Figure 5k). Reflection spectrometry performed on such inverted opals (inset Figure 5i) reveals the existence of five band gaps placed at: *λ*_1_ = 420 nm, *λ*_2_ = 475 nm, *λ*_3_ = 590 nm, *λ*_4_ = 684 nm, and *λ*_5_ = 856 nm. By using Equations (1) and (2) for: *m* = 1; *D* = 264 nm; *f* = 0.74; *n*_1_ = 1 (air refractive index); *n*_2_ = 1.55 (polystyrene refractive index); and *θ* = 0 we can identify the planes family of an fcc lattice that is responsible for the existence of the following: *λ*_1_ (calculated) = 426 nm and corresponds to the (200) planes family; *λ*_2_ (calculated) = 491 nm and corresponds to the (111) planes family; *λ*_3_ (calculated) = 602 nm and corresponds to the (110) planes family; and *λ*_5_ (calculated) = 851 nm and corresponds to the (100) planes family. We failed to attribute a theoretical Bragg band gap correspondence for *λ*_4_ = 684 nm.

The ordered concavity can be used as a unit in the synthesis of higher complexity super-structures, a kind of patterned colloidal crystals in which the units are themselves super-structures. We have thus identified several possibilities. First, bi-dimensional (2D) patterns in which the concavity has an increased complexity, where one or more layers with different refractive indices and the same normal to the concavity surface (represented by N) are superposed (Figure 6a).

In Figure 6b–d, we present three such patterns which have deposited on their inner walls: colloidal Au nanoparticles, polystyrene spheres (0.46 μm in size), and fluorescent silica spheres (0.38 μm in size), respectively.

The 2D high complexity superstructures can also be obtained by infiltrating a binary (20 μm and 50 μm) PS spheres array with silica spheres. In this case, we have the same refractive indices but differently oriented normal directions (Figure 6e). The distribution of the 20 μm and 50 μm spheres in the array is given by the ratio of their mixed volumes (Figure 6f), with the silica spheres possessing an equally good ordering in the different size cavities (Figure 6g). We also point out the existence of a difference in height on the z-axis (Figure 6h) which could generate some interesting optical phenomena and can also be an argument to the observation that during calcination, the silica spheres deposited above the half of the polystyrene spheres are removed. Of course, this approach can be continued with the previous one; in each of the cavities, other materials can be deposited so that in a controlled way both variations of the refractive indices and the normal orientation of the concavity walls or even other physical properties such as new chemical or biological properties of the concavities can be obtained.

Second, the concavity-type unit superstructures can be arranged in three dimensions (3D) so that the orientation of the normal direction at the concavity walls varies in a controlled manner in 3D (Figure 7a).

Superstructures can be integrated into higher complexity superstructures on larger size scales of hundreds of micrometers by using a polyester fabric [34], where polystyrene spheres of tens of micrometers can be forced to follow the pattern of the fabric and to self-assemble on a large area (Figure 7b,c). After infiltration of the silica spheres and the removal of the polystyrene spheres by calcination, higher complexity superstructures (such as those in Figure 7d) composed only of silica spheres are obtained. If instead of calcination (T > 350 °C), the polystyrene/silica grid is only heated to 250 °C for PS melting and infiltration between silica spheres, followed by silica dissolution by HF solution, an inverse opal-like higher complexity superstructure is obtained (Figure 7e). These polystyrene high complexity superstructures appear red in transmission light-optical microscopy (Figure 7f) and have a complex way of reflecting and diffracting white light (Figure 7g). UV-Vis spectrometry (not shown) present the same band gaps structure as that from the Figure 5i inset.

The higher complexity structured colloidal crystal from Figure 7g is only one of the huge number of structures that can be imagined and constructed. By using both large size polystyrene spheres and polyester fabric as templates for silica spheres-direct self-assembly, we can take advantage of their common characteristic, namely, their ability to sustain a sequential infiltration. Even more, the templates’ structural parameters such as: the shape and size of the fabric structural unit; the polystyrene and silica spheres size; and the colloids concentration, or the special physical, chemical or biological qualities of materials which can be added as nanoparticles, thin films, gases and fluids which flow through the pores to the high complexity superstructures allowed us to search for complexity.

## 4. Conclusions

Large size polystyrene spheres can be used as templates for the direct self-assembly of sub-micron silica spheres as ordered concavities that are able to produce the Bragg diffraction of light. Silica spheres form a close packed fcc lattice where light reflection/transmission on (111) and (200) planes families generates the most important bandgaps in the visible domain. The main optical bandgaps of such concavities are produced by the (111) and (200) family planes of an fcc lattice.

Light diffraction on these ordered concavity arrays is weak-angle dependent in the 0–20° range of the incident angle; They present a concavity shaped effect when acting as diffraction rings, and offers a complex way to reflect and diffract white light.

Polystyrene inverted-opal concavity arrays can be fabricated by heating the PS–SiO_2_ binary colloidal crystal at 250 °C, inducing the capillary infiltration from inside of the melted polystyrene that is provided by the large size PS spheres, followed by silica dissolution in aqueous HF.

An ordered concavity can serve as a structural unit in 2D and 3D high complexity superstructure synthesis, in which the orientation of the normal direction on their surface and the light refractive index can be varied in a controlled manner.

Three dimensional-ordered hierarchically porous materials with a huge complexity can be fabricated in a simple, cheap, and fast approach by using large polystyrene spheres as templates for the direct self-assembly of sub-micron silica spheres.

## Figures and Tables

**Figure 1 polymers-14-02158-f001:**
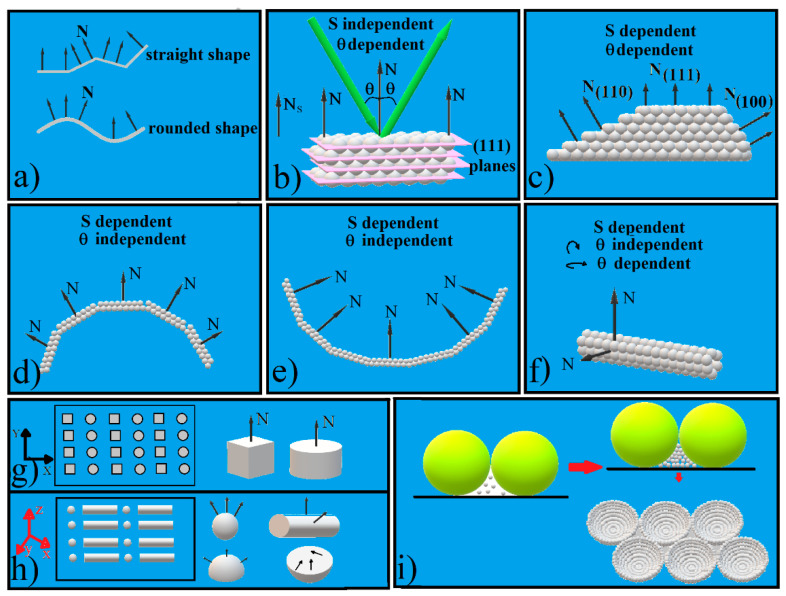
Schematical of: (**a**) straight and rounded shapes; (**b**) flat surfaces parallel with the substrate CC; (**c**) truncated pyramid shaped CC; (**d**) convex, hemispherical shaped CC; (**e**) concave, hemispherical shaped CC; (**f**) cylindrical shaped CC; (**g**) 2D structured patterned CC; (**h**) 3D structured patterned CC; (**i**) ordered concavity array synthesis.

**Figure 2 polymers-14-02158-f002:**
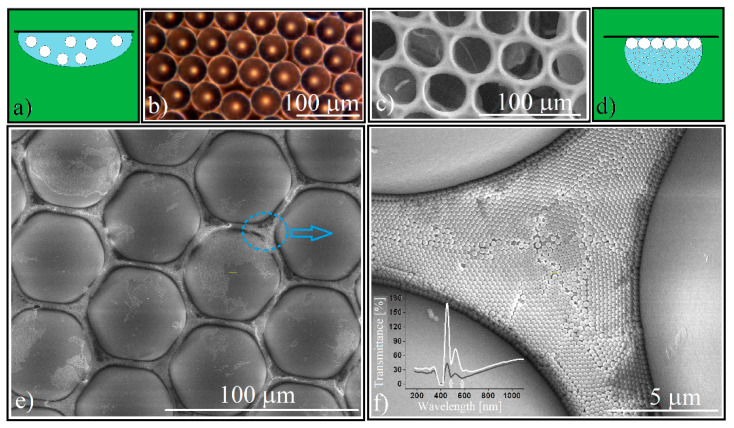
(**a**) Schematical of large size PS spheres single layer synthesis by hanging drop method; (**b**) optical microscopy image of 50 μm PS spheres single layer; (**c**) optical microscopy image of sodium silicate inverse-opal; (**d**) schematical of sub-micron silica colloid infiltration by hanging drop method; (**e**) SEM image of silica spheres self-assembled in the voids between PS spheres; (**f**) SEM image of silica colloidal crystal lattice, inset: transmittance of 50 μm PS and 0.264 μm silica spheres forming a patterned CC (white curve); transmittance of 0.264 μm silica spheres self-assembled as a block rectangular CC, by hanging drop method (black curve).

**Figure 3 polymers-14-02158-f003:**
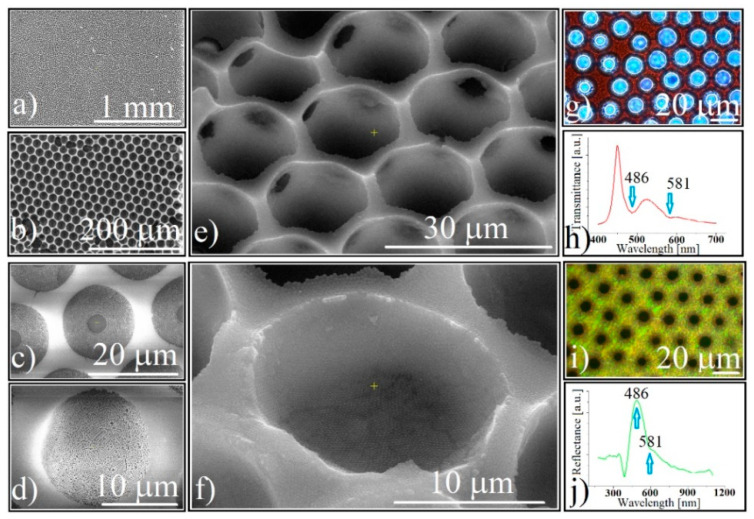
Direct SEM images of: (**a**) large view of silica-ordered concavity array; (**b**) silica-ordered concavity array honeycomb-like structure; (**c**) silica-ordered concavity synthesized by a sequential approach; (**d**) silica-ordered concavity synthesized by co-assembly; (**e**,**f**) tilted (45°) SEM images of silica-ordered concavities; (**g**) transmission optical microscopy image of 0.264 μm silica spheres forming ordered concavities; (**h**) transmittance of 0.264 μm silica spheres forming ordered concavities; (**i**) reflection optical microscopy image of 0.264 μm silica spheres forming ordered concavities; (**j**) reflectance of 0.264 μm silica spheres forming ordered concavities.

**Figure 4 polymers-14-02158-f004:**
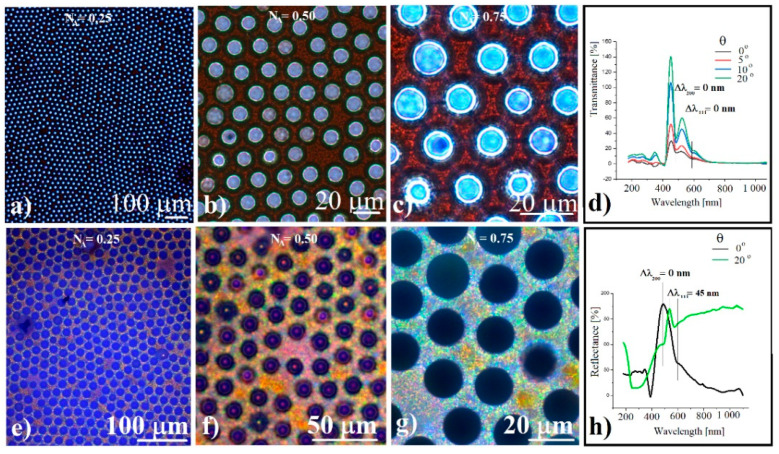
(**a**–**c**) Transmission optical microscopy images of silica spheres concavity array for different focusing and numerical apertures; (**d**) angle-resolved transmittance of silica spheres ordered concavities; (**e**–**g**) reflection optical microscopy images of silica spheres concavity array for different focusing and numerical apertures; (**h**) angle-resolved reflectance of silica spheres ordered concavities.

**Figure 5 polymers-14-02158-f005:**
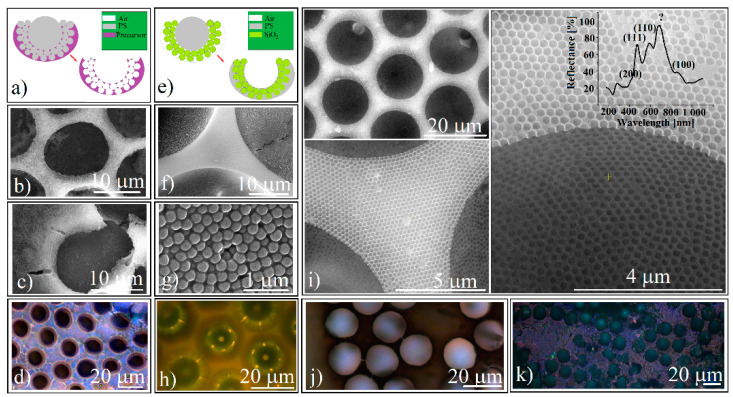
(**a**) Schematical of liquid precursor infiltration and calcination steps for inverse opal-ordered concavities synthesis; (**b**) SEM image of sodium silicate (Na_2_SiO_3_) inverse opal-ordered concavities synthesized by liquid precursor infiltration; (**c**) SEM image of TiO_2_ inverse opal-ordered concavities synthesized by liquid precursor infiltration; (**d**) reflection optical microscopy image of sodium silicate inverse opal-ordered concavities synthesized by liquid precursor infiltration; (**e**) Schematical of polystyrene melt infiltration from inside for PS inverse opal-ordered concavities synthesis; (**f**) SEM image of polystyrene-infiltrated ordered concavities; (**g**) SEM close-up view of polystyrene-infiltrated concavities; (**h**) reflection optical microscopy image of polystyrene infiltrated concavities; (**i**) SEM images of polystyrene inverse opal concavities array at three different magnifications and their reflectance spectrum (inset); (**j**) transmission optical microscopy image of polystyrene inverse opal concavities array; (**k**) reflection optical microscopy image of polystyrene inverse opal concavities.

**Figure 6 polymers-14-02158-f006:**
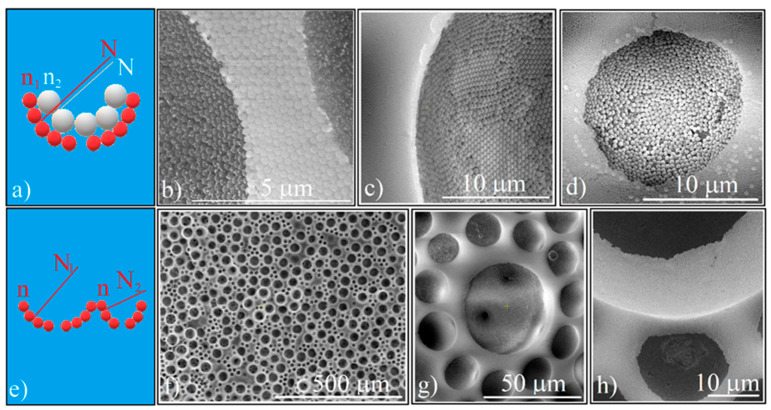
(**a**) Schematical of 2D high complexity superstructures which have an ordered concavity as structural unit (same normal different refractive indices); (**b**) SEM image of Au colloidal particles deposited on the silica ordered concavity wall; (**c**) SEM image of polystyrene (0.488 μm) colloidal particles deposited on the silica ordered concavity wall; (**d**) SEM image of fluorescent silica (0.384 μm) colloidal particles deposited on the silica (0.264 μm) ordered concavity wall (**e**) Schematical of 2D high complexity superstructures which have an ordered concavity as structural unit (different normal, same refractive index); (**f**) SEM image of binary ordered concavity array; (**g**,**h**) SEM images of ordered concavities of different diameter.

**Figure 7 polymers-14-02158-f007:**
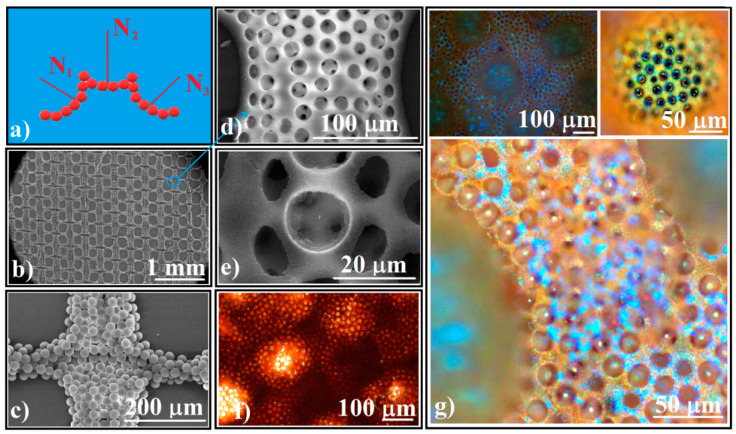
(**a**) Schematical of 3D high complexity superstructures which have an ordered concavity as structural unit; (**b**,**c**) SEM images of 20 μm polystyrene spheres hierarchical self-assembled as a grid; (**d**) close view SEM image of 3D silica spheres high complexity superstructure; (**e**) close view SEM image of 3D polystyrene inverse opal high complexity superstructure; (**f**) transmission optical microscopy image of 3D polystyrene inverse opal high complexity superstructure; (**g**) reflection optical microscopy images of 3D polystyrene inverse opal high complexity superstructure.

## Data Availability

Not applicable.

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
