# Peer review of "Shaping in the Third Direction; Fabrication of Hemispherical Micro-Concavity Array by Using Large Size Polystyrene Spheres as Template for Direct Self-Assembly of Small Size Silica Spheres"

_polymers, 2022, doi:10.3390/polym14112158_

Round 1
Reviewer 1 Report
the article titled "Shaping in the third direction; fabrication of hemispherical micro-concavity array by using large size polystyrene spheres as template for direct self-assembly of small size silica spheres" is well structured and the authors present it in a good way. Each section is complete and the English is good. The conclusions are appropriate and clear as well as each section where the authors argue in an excellent way the methodology used and the results obtained.
I suggest only a revision of the English language.
Author Response
Thank you for the appreciation of our manuscript. The answer to your comment is listed bellow.
- Reviewer: I suggest only a revision of the English language.
Authors: We carefully checked the whole manuscript. The corrections were made and marked with red colour.
Reviewer 2 Report
In this manuscript, the authors developed a novel strategy for the direct self-assembly of silica spheres. The topic is interesting, and the main results have been organized well. It can be published after addressing the following issues.
- This work focused mainly on the fabrication of silica spheres. It is true that they used the PS templates. However, it is no longer a work that has so much to do with polymer. So, is it suitable to publish in the journal of “Polymers”?
- It is necessary for the authors to check the whole manuscript. There are some little mistakes, e.g., UV-Vis (Line 319) or UV vis (Line 313)?
- The height of concavity is an important parameter. It is also significant to show the assembly way of silica spheres. If AFM measurement failed, the authors can tried other measurements. Maybe, 3-D digital microscope is helpful.
- The scale bars in Figure 3g, 3i, 5d, 5h are absent.
- In this work, the authors used transmission UV-Vis as well as reflection UV-Vis spectrums. How about the transmittance and reflectance?
Author Response
Thank you for the appreciation of our work. The answers to your comments are listed bellow.
- Reviewer: This work focused mainly on the fabrication of silica spheres. It is true that they used the PS templates. However, it is no longer a work that has so much to do with polymer. So, is it suitable to publish in the journal of “Polymers”?
Authors: Beside polystyrene spheres as template, one of the final resulted structure was a quality polystyrene photonic crystal as inverted colloidal crystal. This is important because polymeric photonic crystals are less frequently synthesized even if they could show some remarkable properties induced by their polymeric nature such as flexibility and optical transparency. Also, we propose a new approach of the infiltration step, the locally inside infiltration of a polystyrene source. Usually the infiltration take place by dipping the template in a polymeric liquid, which leads to the formation of a polymeric skin over the photonic crystal which strongly diminishes the overall properties of the photonic crystal. This approach may encourage the synthesis of polystyrene photonic crystals for some applications.
- Reviewer: It is necessary for the authors to check the whole manuscript. There are some little mistakes, e.g., UV-Vis (Line 319) or UV vis (Line 313)?
Authors: We check the whole manuscript. It remains only the “UV-Vis”. The corrections were made and marked with red colour.
- Reviewer: The height of concavity is an important parameter. It is also significant to show the assembly way of silica spheres. If AFM measurement failed, the authors can tried other measurements. Maybe, 3-D digital microscope is helpful.
Authors: Unfortunately, we have no access to a 3-D digital microscope. However, the exactly height of concavity value can be determined in an indirect manner from the SEM images and some mathematical relations on the cape of a sphere.
- Reviewer: The scale bars in Figure 3g, 3i, 5d, 5h are absent
Authors: We modified the figures. The scale bars were introduced in Figure 3g, 3i, 5d, and 5h.
- Reviewer: In this work, the authors used transmission UV-Vis as well as reflection UV-Vis spectrums. How about the transmittance and reflectance?
Authors: The text and some of the figure legends were modified. Every time when we refer at a spectrum we use the transmittance and reflectance terms but it remains UV-Vis transmission or reflection when we refer to the method. The modifications were made and marked with red colour.